

# CAST
## Wand And Whisper



**Autorzy**: Jakub Kozanecki · Kacper Kurp · Adam Mikołajczyk · Wiktor Tomczyk

**Opiekun:** dr. inż. Marek Kopel

**Streszczenie**

Celem naszego projektu było stworzenie innowacyjnej pierwszoosobowej gry 3D typu endless, która łączy dynamiczną rozgrywkę z unikalnymi mechanikami sterowania. Gracz porusza się po mapie, stawiając czoła kolejnym falom przeciwników, wykorzystując zaklęcia aktywowane gestami myszy oraz komendami głosowymi. Rozgrywka opiera się na rozwijaniu postaci, odblokowywaniu nowych zaklęć oraz ulepszaniu umiejętności, co pozwala sprostać coraz trudniejszym wyzwaniom.

W wyniku pracy zespołowej zaimplementowaliśmy system rozpoznawania gestów i mowy, który umożliwia immersyjne i intuicyjne sterowanie. Gra została zaprojektowana z myślą o progresji umiejętności oraz utrzymaniu wysokiego poziomu zaangażowania gracza.

Projekt podkreśla potencjał wykorzystania nietypowych interfejsów użytkownika w grach komputerowych, co może znaleźć zastosowanie nie tylko w branży rozrywkowej, ale być może również w edukacji, czy rehabilitacji. Po wstępnych testach nasz sposób sterowania podnosi atrakcyjność rozgrywki oraz zapewnia bardziej immersyjne doświadczenie dla gracza.

## 1 WSTĘP

Współczesne gry komputerowe stale dążą do zwiększania immersji oraz interaktywności, jednak standardowe mechaniki sterowania często ograniczają możliwości graczy. Nasz projekt powstał w odpowiedzi na potrzebę innowacji w tej dziedzinie, proponując mechanizmy sterowania oparte na gestach myszy i komendach głosowych. Problemem, który postanowiliśmy rozwiązać, było zwiększenie zaangażowania gracza i dostarczenie bardziej intuicyjnego doświadczenia w grze 3D typu endless. Kluczowym wyzwaniem było opracowanie systemu, który skutecznie rozpoznaje ruchy i mowę w dynamicznym środowisku gry.

Głównym celem projektu było stworzenie technicznie zaawansowanej gry, która łączy tradycyjne aspekty rozgrywki z innowacyjnymi metodami sterowania. W kontekście biznesowym projekt miał na celu pokazanie potencjału integracji alternatywnych rozwiązań takich jak przetwarzanie gestów i mowy, w celu wyróżnienia się na konkurencyjnym rynku gier, gdzie aktualnie brakuje produktu łączącego te dwie technologie. Technicznie chcieliśmy zademonstrować skuteczność i stabilność takich rozwiązań w warunkach ciągłej i dynamicznej gry.

Zespół stawiał sobie za cel stworzenie gry, która dostarcza nowatorskich doświadczeń i oferuje długotrwałe zaangażowanie graczy dzięki progresji oraz ulepszaniu postaci. Oczekiwane korzyści obejmowały wyższy poziom immersji, możliwość zastosowania opracowanych mechanizmów w innych projektach oraz potencjał przyciągnięcia szerszego grona odbiorców dzięki unikalnym funkcjonalnościom. Projekt stanowi istotny wkład w rozwój interfejsów użytkownika, podkreślając ich znaczenie w nowoczesnej branży rozrywki cyfrowej.

## 2 PRACE NAD PROJEKTEM

W obszarze gier komputerowych istnieją rozwiązania wykorzystujące nietypowe interfejsy użytkownika, takie jak sterowanie głosowe czy oparte na gestach. Przykłady obejmują gry wykorzystujące rozpoznawanie gestów (np. War of Wizards [1]) lub aplikacje integrujące rozpoznawanie mowy (np. Phasmophobia [2]). Jednakże większość gier korzystających z tych technologii ogranicza się do pojedynczych mechanik, rzadko łączac je w spójną całość. Nasz projekt wyróżnia się integracją dwóch komplementarnych systemów sterowania - gestów myszy i komend głosowych - które wspólnie zwiększają interaktywność i immersję.

Główne założenia projektowe obejmowały wybór technologii takich jak silnik Unity 3D, umożliwiający łatwą implementację mechanik rozgrywki i kompatybilność z bibliotekami do rozpoznawania mowy oraz gestów. Zespół pracował z ograniczonym czasem, co wymusiło priorytetyzację funkcji – skupiono się na

podstawowej mechanice walki i progresji, pozostawiając mniej istotne elementy do potencjalnej rozbudowy w przyszłości.

Do realizacji projektu wybraliśmy silnik Unity, który jest wiodącym narzędziem do tworzenia gier 3D. Jego rozbudowane funkcjonalności, intuicyjny interfejs oraz bogata dokumentacja [3] sprawiły, że był on idealnym wyborem dla naszego zespołu. Unity oferuje również liczne wtyczki i integracje, w tym wsparcie dla bibliotek rozpoznawania gestów i mowy, co znacznie przyspieszyło implementację kluczowych mechanik gry.

Do tworzenia modeli postaci, przeciwników i elementów środowiska zdecydowaliśmy się na wykorzystanie Blendera [4]. Narzędzie to jest darmowe, oferuje szeroką gamę funkcjonalności i jest doskonale zintegrowane z Unity, co pozwoliło na płynne importowanie modeli bez utraty jakości. Blender, choć niezwykle potężny, okazał się dla nas niemałym wyzwaniem - większość zespołu miała ograniczone doświadczenie z jego interfejsem i funkcjonalnościami, co wymagało intensywnej nauki.

Ponadto, jednym z największych problemów, z jakimi się zmierzyliśmy był system kontroli wersji. Ogromna liczba konfliktów podczas merge'owania zmian była szczególnie trudna do rozwiązania w przypadku scen, których dane są przechowywane w plikach meta. W celu ich ograniczenia wprowadziliśmy bardziej restrykcyjne zasady współpracy i wyznaczyliśmy osoby odpowiedzialne za zarządzanie zmianami w scenach. Działania te były kluczowe dla utrzymania spójności projektu i sprawnej realizacji założeń.

## 3   WYNIKI

Aby zrealizować wstępne założenia projektu, rozpoczęliśmy od stworzenia podstawowej wersji mapy, wdrażając mechaniki poruszania się gracza oraz opracowując pierwszą, autorską wersję algorytmu rozpoznawania kształtów. Niestety, podczas testów okazało się, że skuteczność tego algorytmu wynosiła jedynie 50%, co zdecydowanie nie spełniało naszych oczekiwań.

Po analizie różnych alternatyw postanowiliśmy wdrożyć algorytm Dollar-P [5]. Algorytm ten służy do rozpoznawania gestów 2D w postaci chmury punktów. Porównuje dwa gesty, próbując znaleźć najlepsze dopasowanie między punktami w obu chmurach, ignorując kolejność, kierunek i liczbę pociągnięć. Został zaprojektowany z naciskiem na dynamiczne środowiska, gdzie przypisania muszą być aktualizowane bardzo szybko w odpowiedzi na zmiany. Podczas testów skuteczność wyniosła około 95%.

Do rozpoznawania komend głosowych, po dokładnej analizie dostępnych technologii, wybraliśmy Windows Speech Recognition. Decyzję tę podjęliśmy ze względu na wbudowaną integrację z silnikiem Unity, szeroką dostępność na komputerach z systemem Windows 7 i nowszymi oraz skuteczne wsparcie dla akcentów nienatywnych użytkowników języka angielskiego. Co więcej, technologia ta zyskała uznanie w konkurencyjnych produktach, takich jak popularna gra „Phasmophobia" [2], gdzie jej skuteczność została potwierdzona w praktyce. Te atuty utwierdziły nas w przekonaniu, że to rozwiązanie idealnie spełni nasze potrzeby.

Gra korzysta z lokalnej bazy danych SQLite [6], która przechowuje wyniki graczy. Dzięki temu wprowadziliśmy element rywalizacji, który nie tylko angażuje użytkowników, ale również zachęca ich do regularnych powrotów. Choć funkcja ta jest stosunkowo prosta, znacząco podnosi atrakcyjność naszej gry i zwiększa jej potencjał rynkowy.

Na Rysunku 1. zaprezentowano mechanizm rysowania kształtów na ekranie, który jest kluczowym elementem interakcji gracza z naszą grą. Efekt końcowy tego procesu, w postaci wybuchu kuli ognia, został ukazany na Rysunku 2. – dynamiczna eksplozja dodaje dramatyzmu i podkreśla moc wykonanego zaklęcia. Z kolei Rysunek 3. przedstawia widok naszej mapy w świetle zachodzącego słońca. Wprowadzenie cyklu dnia i nocy nie tylko urozmaica wizualną stronę rozgrywki, ale również wpływa na atmosferę i immersję gracza. Taki dynamiczny świat pozwala graczowi poczuć się częścią żyjącego i zmieniającego się otoczenia. Od samego początku naszym celem było stworzenie gry, która nie tylko dostarczy rozrywki, ale również zapewni graczom nowatorskie i immersyjne doświadczenia. Po zakończeniu prac nad wersją beta – zawierającą wszystkie kluczowe mechaniki, dopracowane modele oraz efekty dźwiękowe – udostępniliśmy grę do testów grupie ośmiu znajomych. Wyniki były obiecujące: średnia ocena w skali od 0 do 10 wynosiła 8,8. Testerzy szczególnie docenili intuicyjny interfejs użytkownika, który zwiększa immersję, oraz innowacyjny system sterowania.

Najczęściej pojawiającą się uwagą była powtarzalność rozgrywki oraz mało realistyczny świat gry. Problemy te wynikają głównie z ograniczeń czasowych, które uniemożliwiły wprowadzenie bardziej złożonych elementów oraz rozbudowy mechanik. Niemniej jednak, w ciągu ostatnich dwóch tygodni planujemy dodać zaawansowane efekty postprocesingu, które – mamy nadzieję – znacząco poprawią odbiór naszego produktu i uczynią go jeszcze bardziej atrakcyjnym dla graczy.

Choć zdajemy sobie sprawę, że grupa testerów była ograniczona liczebnie i subiektywna, ich pozytywne opinie motywują nas do dalszego doskonalenia projektu. Naszym głównym celem było stworzenie gry możliwie angażującej i immersyjnej, a opinie testerów pozwalają nam sądzić, że udało się go osiągnąć.

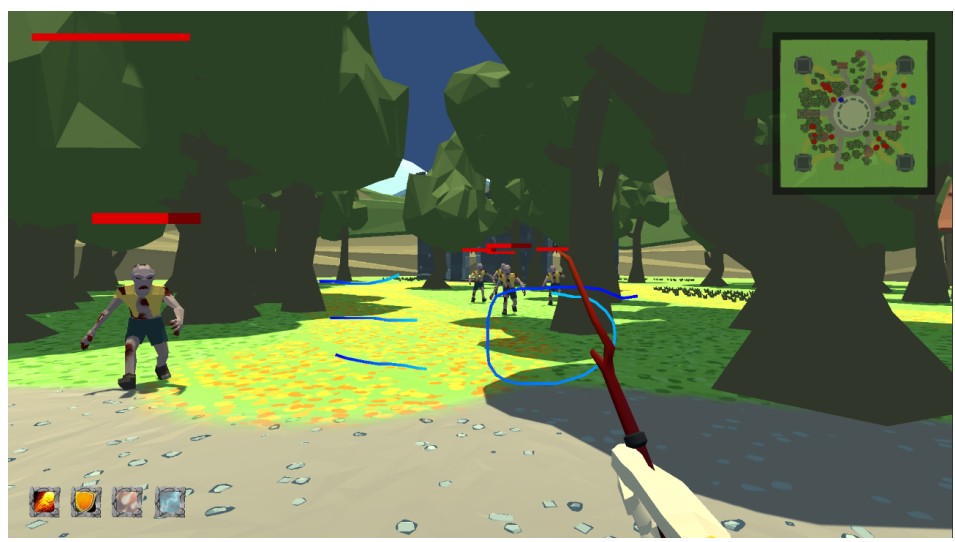

Rysunek 1: Rysowanie kształtów na ekranie

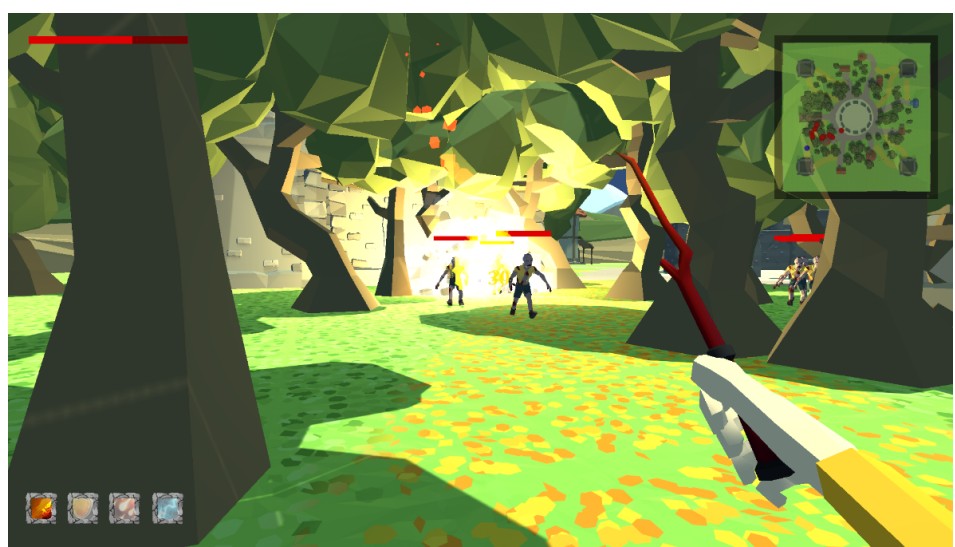

Rysunek 2: Efekt wizualny jednego z zaklęć

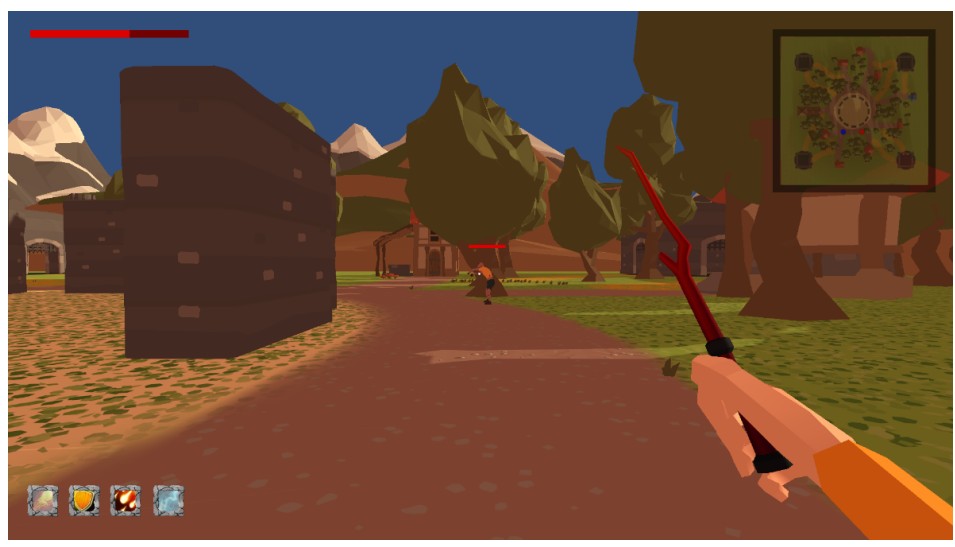

Rysunek 3: Widok sceny przed zachodem słońca

# 4   WNIOSKI

Projekt zakończył się sukcesem, dostarczając innowacyjną grę 3D, która harmonijnie łączy tradycyjne mechaniki rozgrywki z nowoczesnymi metodami sterowania, opartymi na rozpoznawaniu gestów myszy oraz komend głosowych. Osiągnęliśmy kluczowe cele techniczne, w tym wysoką skuteczność algorytmu do rozpoznawania kształtów oraz sprawną integrację systemu rozpoznawania mowy. Dodatkowo, dzięki wprowadzeniu scenek fabularnych, gra zyskała na atrakcyjności, co pozytywnie wpływa na zaangażowanie użytkowników i wydłuża czas spędzony na zabawie.

Wersja beta gry została przetestowana przez ograniczoną grupę użytkowników, którzy ocenili ją bardzo pozytywnie – średnia ocena wyniosła aż 8,8 na 10. Testerzy szczególnie chwalili intuicyjny interfejs użytkownika oraz innowacyjne mechaniki sterowania, które znacząco wyróżniają nasz produkt. Pomimo ograniczeń czasowych, które uniemożliwiły stworzenie bardziej rozbudowanej rozgrywki, projekt spełnił swoje główne założenia, oferując grywalny i immersyjny produkt.

Największym sukcesem projektu jest skuteczna implementacja systemów rozpoznawania gestów i mowy, które zapewniają nowatorskie sterowanie grą. Rozwiązania te mają potencjał do zastosowania nie tylko w grach, ale również w innych aplikacja interaktywnych, co czyni je istotnym krokiem w kierunku rozwoju interfejsów użytkownika. Dzięki temu projekt stanowi cenny wkład zarówno dla branży gier, jak i szeroko pojętego rozwoju technologii interakcji człowiek-komputer. Warto również podkreślić nasz postęp w zakresie modelowania i animacji 3D, które zrealizowaliśmy przy pomocy Blendera. Choć było to dla nas sporym wyzwaniem, efekty naszej pracy okazały się satysfakcjonujące i znacząco podniosły jakość projektu.

# 5   KIERUNKI ROZWOJU

Projekt ma ogromny potencjał do dalszego rozwoju, co pozwala na jego rozszerzenie w wielu interesujących kierunkach. Przede wszystkim, rozgrywka mogłaby zostać wzbogacona o nowe mapy i zadania, które wprowadziłyby większa różnorodność. Funkcjonalnością, gdzie ogranicza nas jedynie wyobraźnia jest drzewko umiejętności. Nowe zaklęcia i ulepszenia można proponować i implementować praktycznie bez końca. Jest to studnia bez dna, a jedynym ograniczeniem jest czas i chęci. Kolejnym krokiem mogłaby być integracja z innymi platformami, takimi jak urządzenia mobilne, gdzie ekran dotykowy posłużyłby do rysowania gestów. Wersja gry na VR pozwoliłaby natomiast graczom w pełni zanurzyć się w wirtualnym świecie, sterując za pomocą ruchów rąk w przestrzeni. Rozbudowa fabularna, w tym wielowątkowy scenariusz z decyzjami gracza wpływającymi na rozwój historii, a także bogatsze dialogi i  cutscenki, zwiększyłyby głębię narracyjną gry.

Optymalizacja gry również stanowi ważny kierunek rozwoju. Analiza danych o rozgrywce mogłaby pomóc w lepszym dostosowaniu poziomu trudności oraz identyfikacji mechanik wymagających poprawy. Jednocześnie wprowadzenie systemu powiadomień zachęciłoby użytkowników do regularnego powracania do gry. Rozwój funkcji multiplayer z dynamicznymi serwerami oraz opcja wsparcia w chmurze umożliwiłyby skalowanie projektu, aby obsłużyć większą liczbę graczy. Dodatkowo, możliwość tworzenia modów przez społeczność otworzyłaby przestrzeń do nieustannego rozwoju gry.

Projekt oferuje praktycznie niezliczone możliwości rozwoju, od wzbogacenia mechanik rozgrywki, przez poprawę immersji, aż po rozszerzenie na nowe platformy i rynki. Tak szeroki wachlarz potencjalnych ulepszeń sprawia, że gra ma szansę stać się nie tylko bardziej zaawansowana technicznie, ale również bardziej angażująca dla graczy. Jako zespół jesteśmy w pełni świadomi potencjału naszego produktu i na pewno nie porzucimy go na obecnym etapie. Po zakończeniu semestru planujemy kontynuować prace nad rozwojem gry, nieustannie wprowadzając nowe funkcjonalności i poprawki. Naszym celem jest dopracowanie jej na tyle, aby w przyszłości móc ją opublikować, udostępniając szerokiemu gronu odbiorców i pokazując efekty naszej ciężkiej pracy.

# LITERATURA

[1] Arcane Miracle Entertainment.  War of wizards, 2022.  Accessed: 2024-11-25.  URL: https://www.warofwizardsvr.com/.

[2] Kinetic Games. Phasmophobia, 2020. Accessed: 2024-11-25. URL: https://www.kineticgames.co.uk/phasmophobia.

[3] Unity Technologies. Unity documentation, 2024. Accessed: 2024-11-26. URL: https://docs.unity.com/.

[4] Blender. Blender, 2024. Accessed: 2024-11-26. URL: https://developer.blender.org/docs/.

[5] DollarP. Dollarp, 2024. Accessed: 2024-11-29. URL: `https://depts.washington.edu/acelab/proj/dollar/pdollar.html`.

[6] SQLite. Sqlite, 2023. Accessed: 2024-11-26. URL: `https://www.sqlite.org/docs.html`.
