# OpenReview forum: "CAST Abstract"
_pwr.edu.pl/Wrocław_University_of_Science_and_Technology/2024/ZPI_Day — Wrocław University of Science and Technology 2024 ZPI Day Submission_

### Official Review · Reviewer_3KCa · 2024-12-05
**Gra: "Harry Potter z myszką zamiast różdżki"**

**Confidence:** 4
**Significance Of Results:** 4
**Overall Quality:** 5

**Compliance With Template:**

5: Very High Quality – The article contains all the required sections, which are written in a very detailed, clear, and error-free manner. The structure is professional and meets expectations, and the content adheres to the highest substantive and formal standards.

**Description Of Results:**

4: High Quality – The results are described in detail and supported by usage examples or evaluations. The description is reliable but may lack full depth of analysis.

**Feedback On Consistency:**

Opis projektu jest spójny i logiczny

**Potential For Development:**

Jaki piszą autorzy: "Projekt oferuje praktycznie niezliczone możliwości rozwoju". Zgadzam się z nimi.

**Project Nature Evaluation:**

Jako grę, trudno ten projekt ocenia w sensie czysto inżynierskim, jednak aspekty techniczne i technologiczne zostały poprawnie "zaadresowane".

**Technical Language Precision:**

5: Very High Quality – The language is entirely appropriate for a technical report. All terms are used correctly and precisely, and the style is professional, clear, and coherent, without any errors or ambiguities.

---

### Official Review · Reviewer_Lnnf · 2024-12-06
**A review of a 3D computer game project that combines speech and gesture recognition for control.**

**Confidence:** 5
**Significance Of Results:** 4
**Overall Quality:** 3

**Compliance With Template:**

3: Average Quality – The article includes most of the required sections, but some may be incomplete, written in a general or unclear manner. The content is correct but requires further refinement.

**Description Of Results:**

4: High Quality – The results are described in detail and supported by usage examples or evaluations. The description is reliable but may lack full depth of analysis.

**Feedback On Consistency:**

The description of the work carried out in terms of the software engineering used is very sparse.

For example, there is no information about the system architecture used or the design patterns applied (if any are applied).

More technical details should be provided – for example, explanation of the synchronization concept of gesture and voice commands applied, information about the testing process and so on.

This creates difficulty in the project assessment.

**Potential For Development:**

The project authors present a number of possible directions for further development work.

**Project Nature Evaluation:**

The project is in its nature a piece of engineering work with development potential, good possible practical application and using interesting algorithmic solutions.

**Technical Language Precision:**

3: Average Quality – The language is mostly appropriate but may contain minor terminological or stylistic errors. Some statements might lack precision or require improvement for better readability.

---

### Official Review · Reviewer_PbTu · 2024-12-06
**Recenzja projektu CAST**

**Confidence:** 5
**Significance Of Results:** 5
**Overall Quality:** 5

**Compliance With Template:**

5: Very High Quality – The article contains all the required sections, which are written in a very detailed, clear, and error-free manner. The structure is professional and meets expectations, and the content adheres to the highest substantive and formal standards.

**Description Of Results:**

5: Very High Quality – The results are described in detail, clearly and comprehensively, supported by thorough evaluation, analysis, and convincing usage examples. The description meets the highest substantive standards.

**Feedback On Consistency:**

Abstrakt napisany jest w sposób spójny i logiczny. Wydaje mi się, że forma bezosobowa byłaby bardziej uzasadniona. Zwróciłam uwagę na sformułowanie "podczas merge'owania zmian" - sugerowałabym unikać form tego typu.
Ponieważ celem było stworzenie gry, zatem subiektywne testy przeprowadzane przez potencjalnych użytkowników są zasadne.
Wyniki zostały przedstawione w sposób właściwy.

**Potential For Development:**

Autorzy zarówno wskazują możliwe kierunki rozwoju, jak i wprost twierdzą, że prace będą kontynuowane. Potencjalnych kierunków rozwoju upatrują zarówno w nowych funkcjonalnościach (umiejętności, zaklęcia, itp.), jak również w zastosowaniu dodatkowych technologii i narzędzi (VR, optymalizacja, itp).

**Project Nature Evaluation:**

Układ pracy jest właściwy, a cel jasno sprecyzowany. Cel został osiągnięty. Uważam, że projekt spełnia wymogi stawiane projektom inżynierskim.

**Technical Language Precision:**

5: Very High Quality – The language is entirely appropriate for a technical report. All terms are used correctly and precisely, and the style is professional, clear, and coherent, without any errors or ambiguities.

---

### Decision · Program_Chairs · 2024-12-10

Accept (Poster)